# Crafting a Foucauldian Archaeology Method: A Critical Analysis of Occupational Therapy Curriculum-as-Discourse, South Africa

**Tania Rauch van der Merwe** [1,2,*] **, Elelwani L. Ramugondo** [3] **and André Keet** [4]

1   Department of Occupational Therapy, School of Therapeutic Sciences University of the Witwatersrand, Johannesburg 2193, South Africa
2   Occupational Therapy Department, University of the Free State, Bloemfontein 9300, South Africa
3   Transformation, Student Affairs and Social Responsiveness, University of Cape Town, Cape Town 7701, South Africa; elelwani.ramugondo@uct.ac.za
4   Rectorate: Engagement and Transformation, Centre for Critical Studies, Nelson Mandela University, Port Elizabeth 6001, South Africa; andre.keet@mandela.ac.za
*   Correspondence: tania.vandermerwe@wits.ac.za

**Abstract:** South Africa has a colonial and apartheid past of social injustice, epistemological oppression, and exclusion. These mechanisms are historically inscribed in the designs, practices, and content of higher education—including in occupational therapy curriculum. If these historical markers are not consciously interrogated, patterns of reproduction are reified along the fault lines that already exist in society. The focus of this article is to demonstrate how an archaeological Foucauldian method was crafted from foundational Foucauldian archaeology analytics and existing approaches of Foucauldian discourse analysis to unearth the rules of the formation of the occupational therapy profession. These rules pertain to the formation of (a) the 'ideal occupational therapist'; (b) who had a say about the profession; (c) the ways of preferred reasoning; and (d) underlying theoretical themes and perspectives about the future. Data sources for this archaeology analytics included commemorative documents of universities on the origin of their programmes; historical regulatory documents; and the South African Journal of Occupational Therapy archive from the period 1953–1994. The analysis rendered two subthemes for each of the rules of formation including 'white exceptionalism', white male national, and international, regulatory bodies, the profession's know-how practical knowledge, and its need for recognition within a bio-medical paradigm. Unearthing the historical markers of a curriculum and viewing it as discourse may enable a conscious reconfiguration thereof.

**Keywords:** critical discourse analysis; Foucauldian archaeology; occupational therapy curriculum

## 1. Introduction

*What I am trying to do is grasp the implicit systems which determine our most familiar behavior without our knowing it. I am trying to find their origin, to show their formation, the constraint they impose upon us; I am therefore trying to place myself at a distance from them and to show how one could escape.* (Foucault in Simon 1971, p. 201)

South Africa (SA) has a colonial and apartheid past of social injustice, epistemological oppression, and exclusion. Universities were largely instrumental in reifying apartheid ideologies and of the 36 higher education institutions, more than half were historically reserved for white people, who were a demographic minority (Badat 2007; Bunting 2006). After SA attained democracy in 1994, radical shifts in higher education were accomplished with a significantly increased number of students that are demographically representative of the country's population accessing higher education, as well as some excellence in areas of teaching and learning, and research (Webbstock 2016; Department of Higher Education, Republic of South Africa 2019). The White Paper on Higher Education and

Training of 1997 (Department of Education, Republic of South Africa 1997) put forward key principles for transformation: equity and redress; democratization; development; quality; effectiveness and efficiency; academic freedom; institutional autonomy; and public accountability. However, almost 30 years after democracy, it appears that Badat's (2007, p. 12) cautioning proved ominous: "... while institutional restructuring is a necessary condition of the transformation in South African higher education it is not a sufficient condition". Student drop-out rates are still over 30% (Statistics South Africa 2017) and white academics constitute 50% of the country's employed academic cohort, regardless of this racial group representing 8% of the country's total population (Wild 2019). The 2015 #Fees Must Fall student movements and protests emerged with force, not only calling for students' financial inclusion but also the decolonization of higher education in South Africa: "... decentering Western epistemologies in higher education curriculum, troubling the alienating nature of university curricula, and the need to transform teaching and learning practices" (Hlatshwayo et al. 2022, p. 2). In a survey conducted by Meyer and Mncayi (2021), 45% of South African graduates are 35 years old and younger, and reported being unemployed against the backdrop of a continuity of previous problems.

It seems then, despite political and structural changes, mechanisms of past social injustice, epistemological oppression, discrimination, and exclusion are historically inscribed in the designs, practices, and content of higher education. Occupational therapy as a profession came into existence in North America and Britain because of the health needs resulting from World War I (Duncan 2017; Friedland 2012; Wilcock 2002), and in South Africa in 1943, at the height of WWII, in the same decade when apartheid was formalized by the Afrikaner Nationalist movement. This historical context begs the following questions: Do current forms of knowledge practices such as the occupational therapy curriculum carry the inscriptions for strengthening the reproductive machineries of departments, and by extension the universities they are part of? To which extent are continuous patterns of unjust inclusion and exclusion part of the function of the curriculum (van der Merwe 2019)?

The occupational therapy profession "came into being because of a basic need and an underlying belief that there is a connection between what people do and their health" (Wilcock 2002, p. 1). Occupational science, a foundational science that emerged from the profession's body of knowledge in the 1980s, also theorized the concept of occupational justice, which is an extension of social justice. Occupational justice underscores the promotion and advocacy for all human beings to choose and engage in daily occupations that enable human dignity, health, and well-being (Duncan and Watson 2004; Hocking 2017; Smith 2017). Townsend (2003, p. 6), one of the founding theorists of occupational justice, explains that occupational injustice occurs when: "participation in daily life occupations is barred, trapped, confined, restricted, prohibited, undeveloped, disrupted, alienated, imbalanced, exploited, deprived, marginalized, or segregated".

However, almost 30 years after SA's democracy, and despite many regulatory imperatives undertaken to break the pattern, in 2018, 66% of occupational therapists who are registered at the Health Professions Council of SA (HPCSA) remain white and mostly female, and the profession continues to be perceived as relatively unknown (Ned et al. 2020). Much critical scholarly work has been performed within a Global South occupational therapy context. Guajardo et al. (2015) urged the profession to critically interrogate its normative assumptions about Global North epistemologies and paradigmatic biases. This critical interrogation is important to move toward contextually responsive reasoning about how people view meaningful and purposeful occupational engagement among a majority of collective-orientated worldviews. Ramugondo (2015, p. 488) argues the imperative concept of occupational consciousness as an "ongoing awareness of the dynamics of hegemony and recognition that dominant practices are sustained through what people do every day, with implications for personal and collective health". Another example of critical theoretical work in Global South occupational therapy is the textbook published by a group of critical occupational therapy scholars titled: "Concepts in Occupational Therapy,

Understanding Southern Perspectives" (Dsouza and Galvaan 2017). On a Global North front, the discourse of prevalent racism and an awareness of its colonial underpinnings in occupational therapy has come to the fore with urgent and renewed calls for more research about the mechanisms of 'discreet' oppressive practices, as well as their historical development and consequences; ongoing and robust critical debate and reflexivity; and problematizing Global North epistemologies and taken-for-granted ways of being (Beagan 2021; Beagan et al. 2022; Emery-Whittington 2021; Johnson et al. 2022; Laliberte Rudman 2013; Lerner and Kim 2022; Sterman and Njelesani 2021).

As an attempted measure of critical deconstruction of historical markers of unjust patterns of inclusion and exclusion, this research was conducted by employing a Foucauldian theoretical toolbox that made a deep understanding of the complex and intricate workings between power, knowledge, and regimes of truth possible. Foucauldian archaeological analytics explain that, when a formal body of knowledge attains the status of science, as occupational therapy did when it was formalized as a programme at universities, it carries with it the markers of the historical discriminators, in terms of, e.g., race, gender, and class. This archaeology analysis was part of a larger study with the overarching question: "How and why does the occupational therapy curriculum, as a politically constructed discourse, create and sustain various patterns of inclusion and exclusion?" (van der Merwe 2019, p. 8). Archaeology aimed to excavate the rules of formation of the occupational therapy profession's implicit knowledge, and therefore its conditions of possibility for the manifestation of a curriculum at a South African university that historically accepted only white students during apartheid. In order to expose, dismantle and transform historical markers of unjust inclusion and exclusion, in-depth archaeology needed to be crafted to excavate the implicit rules of a profession's knowledge formation. Viewing both implicit (savoir) and explicit/formal knowledge (connaissance) in the form of a curriculum, as discourse, offers a critical gaze to identify patterns that must be transformed.

## 2. Materials and Methods

The Foucauldian approach used in this study is situated both as a theoretical approach and crafted method, which is anchored in a critical theory paradigm, including post-structuralist discourse theory. Discourse theory views all social ways of thinking, speaking, doing, and being as meaningful with the premise that these meanings, often tacit, taken-for-granted, and dominant, are constructed from historically situated system/s of rules (Foucault [1970] 1981; Hook 2001; Howarth 2002; Torfing 2005). One example is the way in which madness was constructed between the thirteenth and nineteenth centuries as a social classification and a body of knowledge to be studied. This formation of knowledge was partly because of how any individual or group of people who were deemed as 'unreasonable', or economically unproductive, were sociologically categorized and subsequently confined. However, people who were classified as 'mad' were not only the mentally ill, but also included prostitutes, heretics, or unemployed immigrants unable to speak the local language (Foucault [1961] 1989; O'Farrell 2005; Gregory 2014).

The method of archaeology is understood as a metaphor for excavating the rules of implicit knowledge (Savoir) that created the conditions of possibility for the formal, visible, and explicit knowledge (Connaissance) (Foucault [1969] 2011; Foucault 1998; Schreurich and McKenzie 2006)—in this case, the occupational therapy curriculum (Figure 1). Archaeology is also about making visible the relationships between knowledge and power as "[t]here couldn't be any knowledge without power, and there couldn't be any political power without the possession of a certain special knowledge" (Foucault 2000, p. 31). The purpose of this study was not to "reveal hidden truths" (van der Merwe 2019, p. 40) but rather to show how uninterrogated historical markers may continue to contour and systematically repeat (often unconscious) patterns of inclusion and exclusion as well as its unjust consequences.

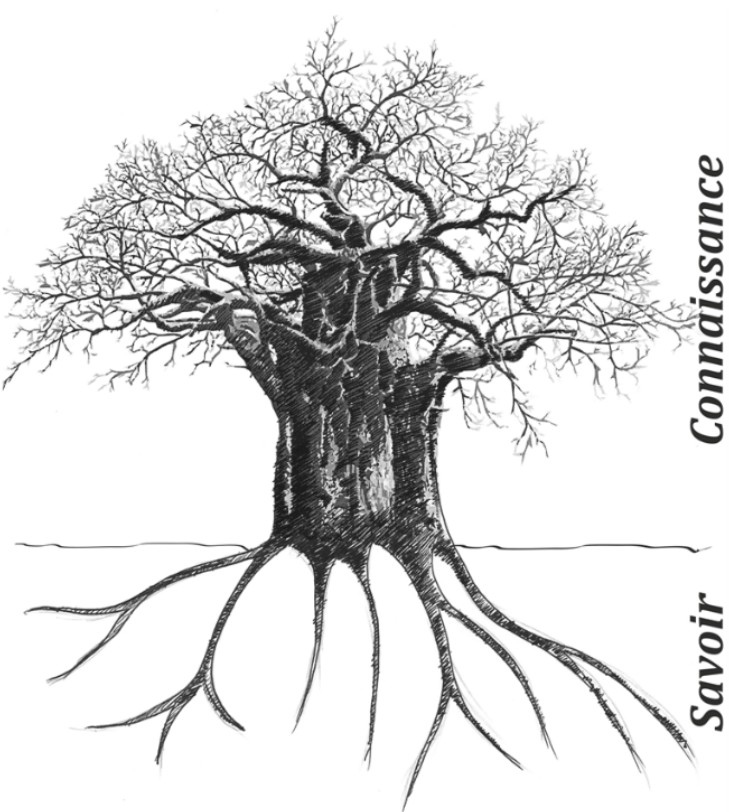

**Figure 1.** Savoir (implicit knowledge) and Connaissance (explicit knowledge) analogically depicted as a Baobab tree.

The first author wanted to stay close to Foucault's original conceptualization of archaeology and wanted to demonstrate a clear distinction, first between the historical knowledge formation of the profession, and then explore how the historical markers thereof may manifest/repeat themselves in a contemporary curriculum. She, therefore, did not want to combine archaeology and genealogy from the outset, as illustrated by the very helpful FDA method put forward by Arribas-Ayllon and Walkerdine (2017). The archaeology method was therefore mainly crafted from Foucault's work (Foucault [1969] 2011) as well as several works of discourse–theory by Foucauldian scholars. Four overarching steps, or rather phases in a process, were followed for the development of the method (Figure 2): (i) First, to understand the theory and concepts of archaeology and their relationships; (ii) Second, identifying and selecting the data, i.e., important speech acts (Dreyfus and Rabinow 1982) which will contain the rules of formation, as well grouping the various sets of data based on shared characteristics (e.g., Sawyer 2002); (iii) Third, analysing the data in terms of their origin, context, and appearance (Jäger and Maier 2009); as well as (iv) Finally, analysing the data in terms of the rules of formation. While the phases of this process are chronologically outlined, they were not necessarily linear but quite re-iterative.

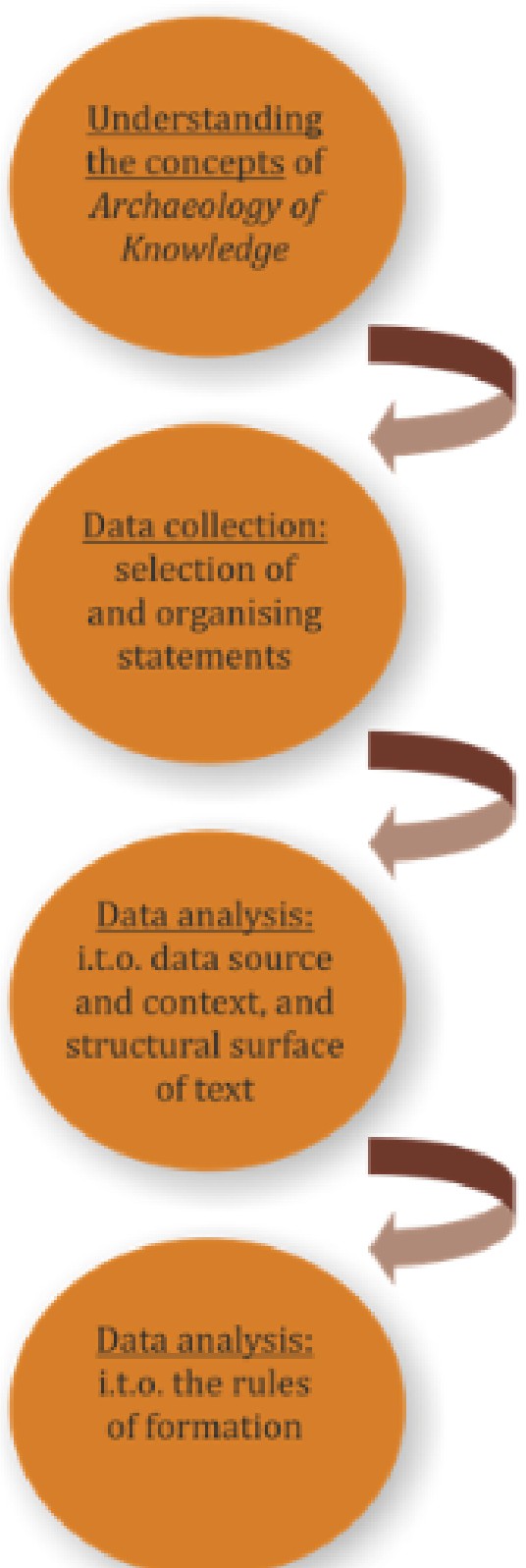

**Figure 2.** An illustration of the process followed in crafting an archaeology FDA.

The first author did extensive reading to fulfil phase (i) and simultaneously conducted multiple literature reviews informed by Foucauldian literature across several disciplines, staying closest to Foucault's Archaeology of Knowledge (Foucault [1969] 2011)[1] with augmented reading for the operationalization of the data collection and analysis. Howarth's (2002) astute analogical outline of Foucauldian concepts as "conceptual machinery" (p. 52) was a main point of departure in this phase, along with Keller's (2013) and Gutting's (1995) clear illuminations of the aspects of Foucauldian archaeology. Jäger and Maier's (2009) interpretation of the practical application of the identification of concepts (ways of preferred reasoning about the profession's knowledge), and strategies (underlying theoretical themes, and perspectives about the future), was also used.

Phase two's (ii) data collection comprised three categories of data. The first set of data was historical documents about how the first occupational therapy students/therapists in South Africa experienced and described the inception of occupational therapy in SA. This set of documents included commemorative texts about the beginning of occupational therapy at universities and the South African Occupational Therapy professional association (SAOT); lecture material that was originally given to students at the specific historically exclusive university about the profession's origin and workings in SA; as well as the transcriptions of two interviews with retired staff members who were part of the beginning of the department at this university.

The second set of data was governing documents formalizing occupational therapy knowledge (a definitive point in legitimizing a fledgling profession to a recognized discursive formation), which included SA government-published documents about the scope and inclusion of the occupational therapy profession as part of the South African Medical and Dental Council; the World Federation of Occupational Therapists (WFOT) "Minimum Educational Standards" (Spackman 1969) as well as its prescribed code of ethics for occupational therapists; and finally, WFOT's chronicle account of the profession from 1952 to 1992.

The third set of data was categorized as historical documents written by academic experts which consisted of the South African Journal of Occupational Therapy archive from 1953 to 1994 (when SA attained democracy), and with an additional focus on the Vona du Toit memorial lectures—a prestige award delivered during the profession's conference, initially annually and later biannually.

Phase (iii) entailed the analysis of the data sources and context, e.g., why the text was important, who the author was, and its historical context. In terms of the appearance (structural surface of the text), an analysis was performed on, e.g., how the content was organized and what symbols or pictures were present in the documents.

Phase (iv) entailed a detailed analysis of the rules of formation, also known as discursive regularities, through thematic induction. The first layer of coding was in relation to the four rules of formation. The second layer of coding was based on several sub-questions under each of the rules of formation as outlined by Foucault ([1969] 2011) and explicated by Gutting (1995). Subcategories and categories were induced from the codes under each of the rules of formation, which in turn constituted the four main themes.

While all four aspects and epistemological standards (e.g., Botma et al. [2010] 2016) of trustworthiness were ensured, an additional reflection on quality criteria in social justice work is to, on the one hand, grapple with the standard of objectivity, and on the other, also constantly wrestle with one's subjectivities and positionality as a researcher. The researcher's (main author) subjectivity formed the foundational drive for doing the research, hopefully appealing to readers to "rethink and reimagine current arrangements" (Fine 2006, p. 98) toward critical praxis.

## 3. Results

Ensuing the four rules of formation as themes, two subthemes emerged from each of the rules of knowledge formation with eight categories rendered from the subthemes.

For this article, illustrated below, is an example of the subthemes from each of the rules of formation/discursive regularities situated in the savoir plane of Figure 1.

### 3.1. Formation of the Ideal Occupational Therapist: White Exceptionalism

The occupational therapy profession was implemented as the first university programme of its kind in South Africa in 1943—five years before apartheid was formalized by the state. Universities were initially grouped according to their implicit political orientation, which manifested firstly by language, but also by race. The Extension of University Act No. 45 of 1959—at the height of the apartheid regime—rendered it illegal to register as a person of colour at certain South African Universities without explicit permission from the Minister of Internal Affairs. An analysis of the appearance (structural surface) of a commemorative document of a university that was historically inclusive of white students only found that white females were hyper-prevalent in historical pictures. In historical lecture material from this university (where the occupational therapy programme was implemented in 1976), an outline, as an orientation lecture, was given to students about the available universities in South Africa offering an occupational therapy degree, as well as their language and race exclusions/inclusion, for example:

> *[Name of Historically Political Conservative University], 1976 offers a four-year degree in B Occupational Therapy. Facilities are offered for white students. The annual intake of students . . . is 20 and the medium of study is Afrikaans.* (Unknown 1977, p. 1)

White exceptionalism was coupled with the selection criteria for occupational therapy students, initially put forward by the World Federation of Occupational Therapists (WFOT) (Spackman 1969) and entrenched as a national pre-requisite for the 'high standards of excellence'. These criteria were furthered at some conservative universities with the imperative of virtue, as one of the interviewees recalls at the historically inclusive university for white people only:

> *The selection criteria [for occupational therapy students] were based on academic achievement [in school] and the students were selected by a panel of male medical academia . . . [These selected students] were very good people, moral and tough . . . they cried if they failed but picked themselves up and went on.* (Interviewee 2 2016, p. 2)

### 3.2. Formation of Legitimate Speakers about the Profession: White Male National, and International, Regulatory Bodies

The analysis of the archive revealed four clusters of legitimate speakers (enunciative modalities) who had a say about the profession. The first was white medical specialists and academics who either advocated for, ensured, and/or regulated the implementation of an occupational therapy curriculum at a university as well as its quality.

The second group of important speakers was two female occupational therapists delegated by the Red Cross from Britain to establish the first occupational therapy programme in 1943 at a historically more politically liberal university with English as the medium of instruction, to meet the needs of injured soldiers as well as increasingly large groups of people being hospitalized because of endemic tuberculosis outbreaks. These two women are described in many historical narrative accounts as stalwarts, known for their tenacity, expert knowledge, and commitment to the profession. For example, the ship they travelled underway to South Africa with textbooks and equipment was torpedoed off the coast of Siera Leone and sunk. Despite having lost all equipment and materials, these women continued making their way to Johannesburg in South Africa, arriving approximately a year later than expected (see, e.g., Wilcock (2002)). An example of how they were described in the lecture material about the history of occupational therapy is provided below:

> *The first qualified occupational therapists came to South Africa from Britain in 1942. They recognized the value of activities currently in use but brought knowledge and discipline to refine their application so that the patients received the treatment specifically designed for their particular needs.* (Unknown 1977, p. 2)

A third group of legitimate speakers about the profession was the World Federation of Occupational Therapists which was founded in 1953 by 10 countries, including South Africa. As an international regulatory body, they established the minimum standards for the education of occupational therapists, which were also later applied as international accreditation criteria for occupational therapy programmes.

The fourth and final group of important speakers having legitimate enunciation about the profession was the South African Medical and Dental Council. This council (breaking up for sentences) vigorously regulated the implementation of occupational therapy programmes in South African universities, requiring, for example, an Education Committee at faculty level to review the content and structure of a proposed programme and curriculum. One of the interviewees participating in such a programme recalled:

> The [content and form] of the curriculum was very much determined by whether occupational therapy [i.e., the programme] was 'registrable' at the Medical and Dental Council. This council consisted of medical practitioners, specialists, dentists and psychiatrists who were males and who were white at that time. (Interviewee 2 2016, p. 4)

*3.3. Formation of Modes of Argumentation and Reasoning about the Knowledge that Is Applied: Know-How Practical Knowledge*

Occupational therapy's ontology is situated both in the bio-medical and social theoretical knowledge paradigms. It, therefore, has an ontologically plural epistemological base. However, historically, it was thrusted into the status of a profession because of its holistic, solution-focused, and practical approach to injured soldiers during the World Wars but without a properly developed scientific base. It only started to develop its research base from the 1960s onward (Wilcock 2002). The South African Journal of Occupational Therapy archive from 1953 until the late 1960s is stringed with publications of innovative and visual depictions of aids for physical disabilities, e.g., a device with detailed sketches on how to enable a person with an upper limb amputation to wash dishes (Swain 1974); an adapted tool for a woman with severe arthritis to insert a bobbin into a sewing machine; and a complex wooden toy adapted to enable a child who had polio to ambulate, facilitating childhood development and play (Best and Peart 1978).

*3.4. Formation of Underlying Theoretical Themes, and Perspectives about the Future: Need for Recognition Withing a Biomedical Paradigm*

Occupational therapy was a new profession that is, on the one hand, revered for its practical wisdom and problem-solving, and on the other, viewed at odds with significant scientific evidence and epistemological standing. As a mostly white female profession, it was continuously seeking validation from the mostly white medical fraternity. A recipient of the prestigious Vona du Toit Memorial Lecture, after returning from a visit to the USA, made the following appeal:

> We should be confident and proud of what we have to offer . . . Occupational therapy is an exciting career, an essential service, and a career of the future . . . OTs in South Africa seem to have accepted a humbly subordinate position in the medical field... Our skills and knowledge have improved but our attitude seems to have remained the same. We accept inferior working conditions and low salaries. We struggle to treat 500 patients on a budget for 50, and our feeble protests subside quickly; we cope somehow . . . We have much to be proud of, to talk about. to write and to share . . . Let us look toward the future with enthusiasm and courage and do the thing in style. (Meyer 1979, p. 7)

**4. Discussion**

Universities in South Africa were a systemic tool for the reification of apartheid and white supremacy ideologies as universities were organized according to political orientation, language, and race. While standards for excellence were an important measure for the legitimization of the profession, it was situated in the rationale of Western exceptionalism and white supremacy (Santos 2014). South Africa was a British colony from 1806 until its

declaration as an independent republic in 1961. However, apartheid was formalized in 1948 until 1994. The organization and distribution of occupational therapy's implicit (savoir) and explicit (connaissance) knowledge, were built on Mignolo's (2009) four pillars of the power matrix: race (authority of white exceptionalism); femaleness supported by patriarchy (gender and sexuality); an emerging healthcare profession (knowledge/subjectivity); and the fact that occupational therapy students selected were supported by parents from a certain class (economy). Virtue ethics, and arguably postures of docility (Foucault 1984), are deeply associated with Calvinist and Puritan values of moralism and perfectionism (van der Westhuizen 2007). However, the individualism associated with virtue ethics (Sherman 2016) as well as the posture of charitability implicit to offering care to people who are perceived as morally deserving, are seldom critically interrogated (van der Merwe 2019).

It can be argued on the one hand that, against the backdrop of patriarchy as the norm at that time, the two women from Britain in their tenacious commitment to the expansion of the profession at the height of WWII, is illustrative of the nature of possibility and restrictions of power relations (Foucault 1978). However, the formation of the occupational therapy profession was also intertwined with the pedagogical nature of institutions such as universities and hospitals (Foucault 1984). This was an important disciplinary strategy (Foucault 1984; Lenoir 1993) to attain legitimacy as a fledgling profession, and to cross the threshold from a discursive practice (a collective notion of ideas) to a discursive formation (Foucault [1969] 2011), i.e., a profession. However, it was implicated that, now, at the time of occupational therapy attaining professional legitimacy, it was also assimilated as part of an established epistemological hierarchy. The analogy of the Victorian trinity came to the fore: the medical doctor as the stern father, the female para-medical occupational therapist as the caring mother, and the child-like patient (Andrews 1999; van der Merwe 2019).

Know-how knowledge is often recognized as a physical product to meet a real-world practical problem and is underscored by the art and skill of innovation and ingenuity (Aristotle[349BC] 1962; Foucault 1998). The early occupational therapy profession's knowledge was situated in a coherentist mode of reasoning that relied on the coherence between the various sets of beliefs situated in both bio-medical and social sciences, as well as an "instrumental efficacy" (Gordon 2000, p. xviii) of its problem-solving approach. This mode of reasoning was at odds with the medical fraternity's foundationalist reasoning, which is discipline-bound and linear (Dancy 2005; Lehrer 2000; Sosa 2011), creating an epistemological inferior view of the profession within a bio-medical paradigm. Lack of critical engagement with the homogeneous forces of one or another epistemology may lead to the reification of historically dominant patterns of thinking and speaking (Foucault 1984), and thwart epistemic freedom and liberation (Ndlovu-Gatsheni 2018; van der Merwe 2019).

One of the paradigmatic roots of occupational therapy knowledge is pragmatism. It implicates a strong association with social virtue: " . . . our edifying construct—occupation—may be understood as a social phenomenon, capable of changing the way that society constructs and reconstructs itself" (Morrison 2016, p. 301). While pragmatism can also be concomitant with an open system of knowledge as a way of reasoning, its origin as adopted by the profession is Eurocentric and therefore individualistically slanted (Sherman 2016). It is also not sufficiently critically interrogated. Pragmatism's shadow side is that it does not necessarily want to deal with theory as an a priori consideration, as its first recourse is the axiom of 'what-works-best', practically. Together with occupational therapists' subverted identity, resulting in docility (Foucault 1984; van der Merwe 2019), this may have resulted in an ongoing struggle of the profession not producing adequate evidence of its efficacy; therefore, its struggle for recognition within the bio-medical paradigm.

## 5. Conclusions

Foucault's (e.g., Foucault ([1969] 2011)) archaeological analytics explain that, at the point where a body of knowledge crosses the threshold from a discursive practice to a discursive formation, meaning when occupational therapy was formalized as a profession by being implemented on the university level as a degree, the historical markers, which were

part of its rules of formation, are inserted into its formal knowledge (Connaissance)—as shown in Figure 1. These historical markers are constructed on the level of the profession's a priori, implicit formation of knowledge (savoir) and were constituted by four groups of rules. These rules of formation are about: (a) how the ideal type of occupational therapist was politically and socio-economically configured; (b) who the legitimate speakers were who had the socio-political power and say about the development and standing of the profession; (c) the preferred ways of reasoning for the profession-in-the-becoming about the application of its knowledge; (d) and how it viewed and negotiated tensions about theoretical themes in its knowledge formation, as well as its future perspectives.

While these rules of the formation of implicit knowledge (savoir) were constituted on an a priori level, they become the conditions of possibility, and therefore constitutive of how formal knowledge is organized, presented, and circulated (Howarth 2002). In this case, the occupational therapy curriculum—carries with it the historical codes for the reification of repeating patterns of (unjust) inclusion and exclusion.

Foucault's archaeology analytics (e.g., Foucault ([1969] 2011)) are very valuable qualitative methodical tools for unearthing historical markers at the level of the initial formation of implicit knowledge. This is, however, not where the analysis should end. The next step would be to look at how taken-for-granted ways of thinking, speaking, doing, and being are often normalized and rationalized at the levels of formalized knowledge, in a curriculum, for example. These ways of rationalization are what Foucault would refer to as technologies of power and are situated in the genealogy of the theoretical toolbox for analysis (e.g., Foucault ([1975] 1977)).

Viewing both a profession and its curriculum as discourse can enable a critical historical deconstruction of curriculum to identify how and why patterns of unjust inclusion and exclusion are reproduced—bringing it to the surface for interrogation, critical dialogue, and change.

**Author Contributions:** Conceptualization, T.R.v.d.M.; A.K. and E.L.R.; methodology, T.R.v.d.M.; investigation, T.R.v.d.M.; formal analysis, T.R.v.d.M.; writing—original draft preparation, T.R.v.d.M.; writing—review and editing, T.R.v.d.M.; A.K. and E.L.R.; visualization, T.R.v.d.M.; supervision, A.K. and E.L.R.; funding acquisition, A.K. All authors have read and agreed to the published version of the manuscript.

**Funding:** The inception of this research was funded by The National Research Fund (NRF) sub-project funding (2012–2015) as part of an overall research project in higher education transformation steered by Ronelle Carolissen, Department of Psychology, Stellenbosch University, South Africa.

**Institutional Review Board Statement:** The study was conducted in accordance with the Declaration of Helsinki, and approved by the Institutional Review Board (or Ethics Committee) of the UNIVERSITY OF THE FREE STATE (protocol code ECUFS 91/2012B approved 20 September 2012) for studies involving humans.

**Informed Consent Statement:** Informed consent was obtained from all participants involved in the study.

**Data Availability Statement:** Data for this part of the study, are hard copies, and available at the archive of the Department of Occupational Therapy, School of Health and Rehabilitation Sciences, University of the Free State (UFS), Bloemfontein, South Africa. Please contact the Head of Department, Azette Swanepoel at swanepoela@ufs.ac.za or (+27)51 401 2829 for enquiries. The South African Journal of Occupational Therapy archive is available at the UFS Frik Scott Medical Library. Please contact services at https://ufs.libguides.com/medicallibrary (accessed on 1 May 2023) for enquiries.

**Acknowledgments:** The authors wish to thank the anonymous reviewers for their comments. The main author also wishes to acknowledge the co-authors for their roles as supervisors of the whole of the Ph.D. study.

**Conflicts of Interest:** The authors declare no conflict of interest. The funders had no role in the design of the study; in the collection, analyses, or interpretation of data; in the writing of the manuscript; or in the decision to publish the results.

## Note

[1]    Bearing in mind that many mistranslations of Foucault's work from French to English exist (O'Farrell 2005), the main author also read widely on contestations of his work, e.g., Sawyer's (2002) argument about the commonly decontextualized interpretation of the concept 'discourse' in a much broader sense as what Foucault meant when taking into account the surrounding text in the paragraph, as well as later references about this term in "Archeaology of Knowledge" (Foucault [1969] 2011; van der Merwe 2019).

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
