# Peer review of "Crafting a Foucauldian Archaeology Method: A Critical Analysis of Occupational Therapy Curriculum-as-Discourse, South Africa"

_socsci, doi:10.3390/socsci12070393_

Round 1

Reviewer 1 Report

Thank you for the opportunity to review this piece. I appreciated your methodological approaches and novel interpretation of curriculum as discourse. I am excited to consider this method in my own work focusing on Indigenous college students. Overall, this was an exceptionally written article with no grammatical errors. Great job! 

Reviewer 2 Report

This paper is very informative and illuminating, especially with respect of showing a valuable application of insights derived from Foucault. It could benefit from two improvements:  1) the title mentions neither occupational theory nor the context of South Africa, while both are critical to the paper; and 2) the Conclusion is disappointingly short and could benefit from a more developed argument for future research, qualitative methods, our understanding of Foucault's methods, and so on.

Reviewer 3 Report

This is a very well-written paper that focuses on an important theme of interrogating issues of colonial and apartheid injustice in South African within the rules of the formation of the occupational therapy profession, using an archaeological Foucauldian analysis.

The abstract clearly outlines the main focus of the paper.

The introduction is coherent and comprehensive with multiple links to appropriate sources that effectively support the points made. Yet, I am not sure about the use of the phrase “Foucauldian theory”, as Foucault himself refused to refer to his ideas as a theory, and actually, the word “theory” is completely absent from his work. When Foucault offers his detailed studies on knowledge, power, and forms of social control, he refers to his ideas as to not “theory”, but “toolbox” or “analytics”, insisting that the former are neither a theory nor a methodology. With this in mind, it would have been useful for the authors to provide justification of their usage of the word “theory” in relation to Foucault’s work.

The section on materials and methods provides a broad and thorough discussion of the data collection process and consequent analysis, using the method of archaeology understood as a method for “excavating rules of implicit knowledge”. This, in my view, represents quite an innovative approach to data analysis.

Results and discussion sections are equally comprehensive, providing a good overview of the emerged themes with systematic theoretical analysis, demonstrating how universities in South Africa were a systemic tool for the reification of apartheid and white supremacy ideologies. Perhaps, it would have been useful to make more references in these sections to the actual Foucault’s work, considering the intention of the paper to provide a critical analysis using Foucauldian approach. I felt that this theme has been slightly lost in the discussion.

 Apart from these minor points, in my view, this paper makes a valuable contribution to the existing scholarship on the relevant topics.
